

# Modeling the effects of cyclodextrin on intracellular membrane vesicles from Cos-7 cells prepared by sonication and carbonate treatment

Peter Kilbride[1], Holly J. Woodward[1], Kuan Boone Tan[2],
Nguyễn T.K. Thanh[2], K.M. Emily Chu[1], Shane Minogue[1] and
Mark G. Waugh[1]

[1] UCL Institute for Liver & Digestive Health, University College London, London,
United Kingdom
[2] Biophysics Group, Department of Physics & Astronomy, University College London, London,
United Kingdom

## ABSTRACT

Cholesterol has important functions in the organization of membrane structure and this may be mediated via the formation of cholesterol-rich, liquid-ordered membrane microdomains often referred to as lipid rafts. Methyl-beta-cyclodextrin (cyclodextrin) is commonly used in cell biology studies to extract cholesterol and therefore disrupt lipid rafts. However, in this study we reassessed this experimental strategy and investigated the effects of cyclodextrin on the physical properties of sonicated and carbonate-treated intracellular membrane vesicles isolated from Cos-7 fibroblasts. We treated these membranes, which mainly originate from the *trans*-Golgi network and endosomes, with cyclodextrin and measured the effects on their equilibrium buoyant density, protein content, represented by the palmitoylated protein phosphatidylinositol 4-kinase type II$\alpha$, and cholesterol. Despite the reduction in mass stemming from cholesterol removal, the vesicles became denser, indicating a possible large volumetric decrease, and this was confirmed by measurements of hydrodynamic vesicle size. Subsequent mathematical analyses demonstrated that only half of this change in membrane size was attributable to cholesterol loss. Hence, the non-selective desorption properties of cyclodextrin are also involved in membrane size and density changes. These findings may have implications for preceding studies that interpreted cyclodextrin-induced changes to membrane biochemistry in the context of lipid raft disruption without taking into account our finding that cyclodextrin treatment also reduces membrane size.

Corresponding author
Mark G. Waugh,
m.waugh@ucl.ac.uk

## INTRODUCTION

In this study we investigated the relationship between membrane composition, density, and size by using methyl-$\beta$-cyclodextrin (cyclodextrin) to rapidly deplete membrane cholesterol from an isolated intracellular membrane preparation. Cyclodextrins are a family of cyclic oligosaccharides that adopt a cone-like structure in aqueous solution, with an internal hydrophobic core that can sequester lipids from membranes (*Heine et al., 2007*;

*Pinjari, Joshi & Gejji, 2006*). Cyclodextrins have useful pharmaceutical applications as soluble carriers for hydrophobic molecules and are also commonly used in biochemical and cell biology studies to manipulate membrane lipid levels (*Loftsson & Brewster, 1996*; *Rodal et al., 1999*; *Welliver, 2006*; *Zidovetzki & Levitan, 2007*). Cyclodextrin efficaciously removes sterols such as cholesterol from biological membranes but can also remove other lipids such as sphingomyelin and phosphatidylcholine (*Ottico et al., 2003*). Recently cyclodextrin and the related molecule hydroxypropyl-$\beta$-cyclodextrin have been shown to alleviate the pathological intracellular accumulation of free cholesterol in Niemann-Pick Type C disease models (*Camargo et al., 2001*; *Davidson et al., 2009*; *Holtta-Vuori et al., 2002*; *Lim et al., 2006*; *Liu et al., 2008*; *Liu et al., 2010*; *Liu et al., 2009*; *Mbua et al., 2013*; *Pontikis et al., 2013*; *Ramirez et al., 0000*; *Ramirez et al., 2010*; *Rosenbaum et al., 2010*; *Swaroop et al., 2012*; *Te Vruchte et al., 2014*; *Vance & Karten, 2014*; *Vite et al., 2015*; *Waugh, 2015*). These recent developments demonstrate a potential therapeutic use for cyclodextrins and also clearly establish their efficacy for reducing the cholesterol content of endosomal membranes (*Rosenbaum et al., 2010*; *Shogomori & Futerman, 2001*). In addition, we have previously reported that the addition of cyclodextrin to cultured cells leads to the vesicularization and contraction of the *trans*-Golgi network (TGN) and endosomal membranes (*Minogue et al., 2010*). These cyclodextrin-induced changes to intracellular biomembrane architecture are associated with alterations to intramembrane lateral diffusion and lipid kinase activity of phosphatidylinositol 4-kinase II$\alpha$ (PI4KII$\alpha$), a constitutively palmitoylated and membrane-associated enzyme (*Barylko et al., 2009*; *Lu et al., 2012*) that may be important in the etiology of some cancers and neurodegenerative disorders (*Chu et al., 2010*; *Clayton, Minogue & Waugh, 2013a*; *Li et al., 2010*; *Li et al., 2014*; *Simons et al., 2009*; *Waugh, 2012*; *Waugh, 2014*; *Waugh, 2015*).

Whilst cyclodextrin has been mainly used to remove cholesterol from the plasma membrane our focus here is on characterizing the effects of such treatment on intracellular membranes where cholesterol levels are known to be important for processes such as protein sorting and trafficking from the TGN (*Paladino et al., 2014*). Since the effects of cyclodextrin on intracellular membranes are important to understand both in a disease context (*Vite et al., 2015*) and for furthering our knowledge about the functions of cholesterol on intracellular membranes, we decided to investigate more comprehensively how cyclodextrin alters the biophysical properties of a lipid-raft-enriched membrane fraction isolated from intracellular TGN and endosomal membranes (*Minogue et al., 2010*; *Waugh et al., 2011a*). In particular, we sought to understand more fully the cyclodextrin-induced changes to the equilibrium buoyant densities of isolated cholesterol-rich membrane fractions that we and others have reported in a number of preceding publications (for examples, see *Hill, An & Johnson, 2002*; *Kabouridis et al., 2000*; *Matarazzo et al., 2012*; *Minogue et al., 2010*; *Navratil et al., 2003*; *Pike & Miller, 1998*; *Spisni et al., 2001*; *Xu et al., 2006*; *Zidovetzki & Levitan, 2007*). In these previous experiments, cholesterol depletion with cyclodextrin rendered the membrane fraction less buoyant, leading to the cyclodextrin-treated membranes banding to a denser region in an equilibrium density gradient. This cyclodextrin-induced change, sometimes referred to as a density shift, has allowed us to design, using sucrose density gradients, a membrane floatation assay in

which we have been able to separate cholesterol-replete and -depleted membranes before and after cyclodextrin treatment.

In many of these prior studies, a cyclodextrin-dependent redistribution of biomolecules to a denser membrane fraction was interpreted as a delocalization from cholesterol-rich lipid rafts or liquid-ordered domains to a less buoyant, liquid-disordered, non-raft fraction. This reasoning stems from the idea that raft-enriched membrane domains are intrinsically buoyant due to their high lipid-to-protein ratio. However, since density is defined as mass divided by volume we reassessed these inferences on the grounds that in the absence of a membrane volume change, a reduction in mass alone would result in a membrane becoming more buoyant, i.e., less dense.

To explore the relationship between cholesterol content and membrane density, we employed our membrane floatation assay to measure the change in the physical properties and biochemical composition of cholesterol-enriched membrane vesicles following cyclodextrin treatment. We then analyzed these changes to mathematically model, from first principles, the degree to which the mass and volume of the membrane domains would have to alter to account for the measured change in membrane density. Finally, we provide a mathematical solution to explain the relationship between membrane cholesterol mass and vesicle density.

## MATERIALS AND METHODS

### Materials

All cell culture materials, enhanced chemiluminescence (ECL) reagents and X-ray film were purchased from GE Healthcare Life Sciences, UK. Polyvinylidene difluoride (PVDF) membrane was bought from Merck Millipore UK. Horseradish peroxidase (HRP)-linked secondary antibodies were purchased from Cell Signalling Technology UK. The antibody to PI4KIIα was previously described by us (*Minogue et al., 2010*). HRP-linked cholera toxin B subunit was purchased from Sigma-Aldrich UK. Sucrose was obtained from VWR International Ltd UK. Complete protease inhibitor tablets were purchased from Roche Ltd UK. All other reagents were from Sigma-Aldrich UK.

### Cell culture

Cos-7 cells obtained from the European Collection of Cell Cultures operated by Public Health England were maintained at 37 °C in a humidified incubator at 10% $CO_2$. Cells were cultured in Dulbecco's Modified Eagle's Medium (DMEM) supplemented with Glutamax, 10% fetal calf serum, 50 i.u./mL penicillin, and 50 μg/mL streptomycin. Cell monolayers were grown to confluency in 10 cm tissue culture dishes. Typically, four confluent plates of cells were used in each subcellular fractionation experiment.

### Subcellular fractionation by sucrose density gradient centrifugation

A buoyant subcellular fraction enriched for TGN and endosomal membranes was prepared according to our previously published method (*Minogue et al., 2010*; *Waugh et al., 2006*). Confluent cell monolayers were washed twice in ice-cold phosphate-buffered saline

(PBS) pH 7.4 and then scraped into 2 mL of homogenization buffer (Tris-HCl 10 mM, EGTA 1 mM, EDTA 1 mM, sucrose 250 mM, plus Complete$^{TM}$ protease inhibitors, pH 7.4). Post-nuclear supernatants were prepared by Dounce homogenization of the cells suspended in homogenization buffer followed by centrifugation at 1,000 g at 4 °C for 2 min to pellet nuclei and unbroken cells. Cellular organelles were separated by equilibrium density gradient centrifugation by overnight ultracentrifugation on a 12 mL, 10–40% w/v sucrose density gradient as previously described (*Waugh et al., 2003a*; *Waugh et al., 2003b*; *Waugh et al., 2006*). Using this procedure, a buoyant TGN-endosomal enriched membrane fraction consistently banded in gradient fractions 9 and 10 and was harvested as described before (*Waugh et al., 2003b*; *Waugh et al., 2006*).

## Refractometry to measure membrane density

The refractive index of each membrane fraction was determined using a Leica AR200 digital refractometer. Refractive index values were then converted to sucrose densities using Blix tables (*Dawson et al., 1986*) and linear regression carried out using GraphPad Prism software.

## Membrane floatation assay to measure the equilibrium buoyant density of membrane vesicles

This assay was previously described by us (*Minogue et al., 2010*). Briefly, 400 μL of cyclodextrin (20 mM) dissolved in water was added to an equal volume of TGN/endosomal membranes on ice for 10 min to give a cyclodextrin concentration of 10 mM. Then, 200 μL of sodium carbonate (0.5 M, pH 11.0) was added to a final concentration of 50 mM in a 1 mL sample. The carbonate-treated membranes were probe-sonicated on ice using a VibraCell probe sonicator from Sonics & Materials Inc., USA at amplitude setting 40 in pulsed mode for 3 × 2 s bursts. To the 1 mL sonicated membrane samples, 3 mL of 53% w/v sucrose in Tris-HCl 10 mM, EDTA 1 mM and EGTA 1 mM, pH 7.4 was added to form 4 mL of sample in 40% w/v sucrose and a sodium carbonate concentration of 12.5 mM and, where applicable, a cyclodextrin concentration of 2 mM. A discontinuous sucrose gradient was formed in a 12 mL polycarbonate tube by overlaying the 40% sucrose layer with 4 mL 35% w/v and 4 mL 5% w/v sucrose in Tris-HCl 10 mM, EDTA 1 mM and EGTA 1 mM, pH 7.4. The gradient was centrifuged overnight at 185,000 g at 4 °C in a Beckman LE-80K ultracentrifuge and 12 × 1 mL fractions were harvested beginning at the top of the tube.

## Immunoblotting of sucrose density gradient fractions

The protein content of equal volume aliquots of each density gradient fraction was separated by sodium dodecyl sulfate-polyacrylamide gel electrophoresis (SDS-PAGE), transferred to PVDF membranes and probed with antibodies directed against proteins of interest. Western blots were visualized by chemiluminescence and bands were quantified from scanned X-ray films using image analysis software in Adobe Photoshop CS4.

## Measurements of membrane lipid levels

The cholesterol content of equal volume membrane fractions was assayed using the Amplex red cholesterol assay kit (Molecular Probes). The use of this assay to measure
membrane cholesterol mass has been previously validated (*Bate, Tayebi & Williams, 2008*; *Minogue et al., 2010*; *Nicholson & Ferreira, 2009*). Ganglioside glycosphingolipids were detected by dot blotting of membrane fractions (*Ilangumaran et al., 1996*) and probing with HRP-conjugated cholera toxin B subunit as described previously (*Ilangumaran et al., 1996*; *Mazzone et al., 2006*; *Waugh, 2013*; *Waugh et al., 2011a*; *Waugh et al., 2011b*). Membrane-bound cholera toxin was visualized by incubation with chemiluminescence detection reagents and spots were quantified as described for the analysis of immunoblotting data (*Waugh, 2013*).

## Dynamic light scattering measurement to measure hydrodynamic diameter of membrane vesicles

The hydrodynamic size of the membrane vesicles in the gradient fraction was studied with a Zetasizer Nano ZS90 (Malvern Instruments). All diluted samples were prepared in filtered (0.2 μm) Millipore ddH$_2$O to avoid sample artifacts, and measurements were made at 25 °C in triplicate.

## Mathematical modelling of membrane compositional changes

Subscripts are used to specify a unit being examined, with $s$ and $r$ defining treatment sensitive (assuming that most of this fraction is composed of cholesterol with a density of around $\rho = 1.067$ (*Haynes, 2013*)) and remaining components, respectively. The subscript—post is used to denote values for vesicles post cyclodextrin treatment.

The fractions are not considered discrete sections of the vesicles; rather they can be mixed and inter-connected.

The mass density of a particle is defined as the mass per unit volume. To determine the mass density of an object consisting of multiple discrete components in a steady state, a linear combination of its components can be used as in Eq. (1).

$$\rho = \frac{m_1 + m_2 + m_3 + \cdots}{V_1 + V_2 + V_3 + \cdots} \tag{1}$$

where the subscripts denote the mass and volume of the separate components. Through normalizing the total volume $V = V_1 + V_2 + V_3 + \cdots = 1$, the density can simplify to

$$\rho = m_1 + m_2 + m_3 + \cdots m_n$$

where $m_n$ now refers to the mass of the volume fraction in question. Considering an object composed of $n$ different materials the overall mass density is therefore

$$\rho_{\text{whole}} = \rho_{\text{fraction1}} V_{\text{fraction1}} + \rho_{\text{fraction2}} V_{\text{fraction2}} + \rho_{\text{fraction3}} V_{\text{fraction3}} + \cdots$$

$$= \sum_{j=1}^{j=n} \rho_j V_j \tag{2}$$

where $\rho_j$ is the density of fraction $j$, and $v_j$ is the volume fraction of material $j$.

To determine the % composition of the vesicles, boundary conditions were formulated and solved using simultaneous equations. Pre-treatment, the system was described

through Eq. (3):

$$V_r \times \rho_r + V_s \times \rho_s = \rho_{\text{pre}} \tag{3}$$

where the fractional volume of the residual component is given by $V_r$, the fractional volume of the treatment sensitive component $= V_s$, and $\rho_{\text{pre}}$ was the measured density of the vesicle pre-treatment.

A second simultaneous equation arises through the physical definition of the system, which is the total volume has been normalised to one:

$$V_r + V_s = 1 \tag{4}$$

i.e., combining all the fractions in a vesicle together equaled fraction one of a vesicle.

The 3rd simultaneous equation was determined with respect to the post-treatment density. It can be derived that for the cyclodextrin sensitive fraction:

$$\rho_{\text{pre}} = \frac{m_{\text{pre}}}{V_{\text{pre}}} = \rho_{\text{post}} = \frac{m_{\text{post}}}{V_{\text{post}}} = 1.067. \tag{5}$$

While the mass and volume of the cholesterol fraction change, its intrinsic density does not. Hence:

$$\frac{m_{\text{pre}}}{V_{\text{pre}}} = \frac{m_{\text{post}}}{V_{\text{post}}} \Rightarrow V_{\text{post}} = \frac{m_{\text{post}}}{m_{\text{pre}}} \times V_{\text{pre}} \tag{6}$$

$\frac{m_{\text{post}}}{m_{\text{pre}}} =$ the mass post-treatment relative to the pre-treatment mass, which was defined as the dimensionless parameter $M$. The RHS of Eq. (6) then simplified to: $V_{s\text{-pre}} \times M$. A similar procedure was followed for $V_r$.

In order to define the mass density of the vesicles post-treatment, Eq. (3) was modified and normalized to take account of the change of mass. This yielded:

$$\rho_{\text{post}} = \frac{V_r \times \rho_r + V_s \times \rho_s \times M_{s\text{-post}}}{V_r + V_s \times M_{s\text{-post}}}. \tag{7}$$

## Statistical analysis

Data are presented as mean $\pm$ SEM of at least three determinations. Statistical significance was assessed using the two-tailed student $t$ test and $P$ values $< 0.05$ were deemed to be statistically significant.

## RESULTS

### Changes in membrane composition and density following cholesterol depletion

The starting material for this set of experiments was our previously characterized cholesterol-rich intracellular membrane fraction prepared from post-nuclear cell supernatants. These membranes were isolated on equilibrium sucrose density gradients and their identity as a TGN-endosomal fraction was confirmed by Western blotting for

PI4KIIα and syntaxin-6 (*Minogue et al., 2010*; *Waugh et al., 2006*). To investigate in more detail the relationship between cholesterol levels, membrane composition and membrane biophysical properties, we employed our recently described floatation assay method to determine the equilibrium buoyant densities of TGN-endosomal membrane domains using sucrose gradients (see work flow chart in Fig. 1). This technique involved treating the membranes with cyclodextrin (10 mM) for 10 min to extract cholesterol followed by probe sonication to induce their vesicularization and fragmentation (*Waugh, Lawson & Hsuan, 1999*; *Waugh et al., 1998*). The sonication step was carried out in alkaline carbonate buffer which is a well-established means for removing peripheral proteins including actin from membranes (*Fujiki et al., 1982*; *Nebl et al., 2002*). This procedure was necessary in light of the extensive literature demonstrating that peripherally associated membrane proteins can influence membrane architecture, geometry and density, and such additional heterogeneity in these membrane characteristics could potentially complicate subsequent biophysical analyses. This combination of probe sonication and carbonate addition was aimed at generating a population of membrane vesicles stripped of peripheral proteins including cyclodextrin-sensitive cytoskeletal proteins which have the potential to modify membrane microdomain stability. Furthermore, the inclusion of these treatments meant that the integral protein and lipid compositions of the vesicles would be the principal determinants of membrane density.

In this set of experiments we compared the effects of cyclodextrin treatment on the biochemical composition of the buoyant (fractions 5–8) and dense (fractions 9–12) regions of the sucrose gradient. Cyclodextrin addition resulted in a large ($83.4 \pm 2.75\%$, $n = 3$) decrease in the cholesterol mass of the buoyant fractions protein without any significant accumulation in the denser region of the sucrose gradient (Fig. 2). This large reduction in cholesterol also coincided with a relocalization of the membrane-associated PI4KIIα protein to denser membrane fractions 9–12 (Fig. 2). We quantified this change in PI4KIIα distribution, which was also noted in our previous publication (*Minogue et al., 2010*), and found that unlike the situation with cholesterol, cyclodextrin did not result in an overall loss of PI4KIIα from the membrane fractions.

We used refractometry to measure the sucrose density of the gradient fractions. Trial experiments revealed that the final, diluted cyclodextrin concentrations of 200 mM present in the dense gradient fractions did not impact on the refractive index readings for these samples. These measurements allowed us to determine that the inclusion of cyclodextrin caused the main protein fraction to shift in density from 1.096 to 1.126 g/mL (Fig. 3).

Finally, we measured cyclodextrin-effected changes to the hydrodynamic diameter of the vesicles by dynamic light scattering. Even though the isolated membrane vesicles were found to be heterogeneous we focused on a peak signal corresponding to a vesicle population in the biologically relevant size range of 10–1,000 nm. We ascertained that while there was no change in the total number of vesicles, the average vesicle diameter shrunk from 780 to 42 nm in the buoyant fraction and from 453 to 271 nm in the dense fraction (Table 1). These results showed that the reduction in cholesterol levels

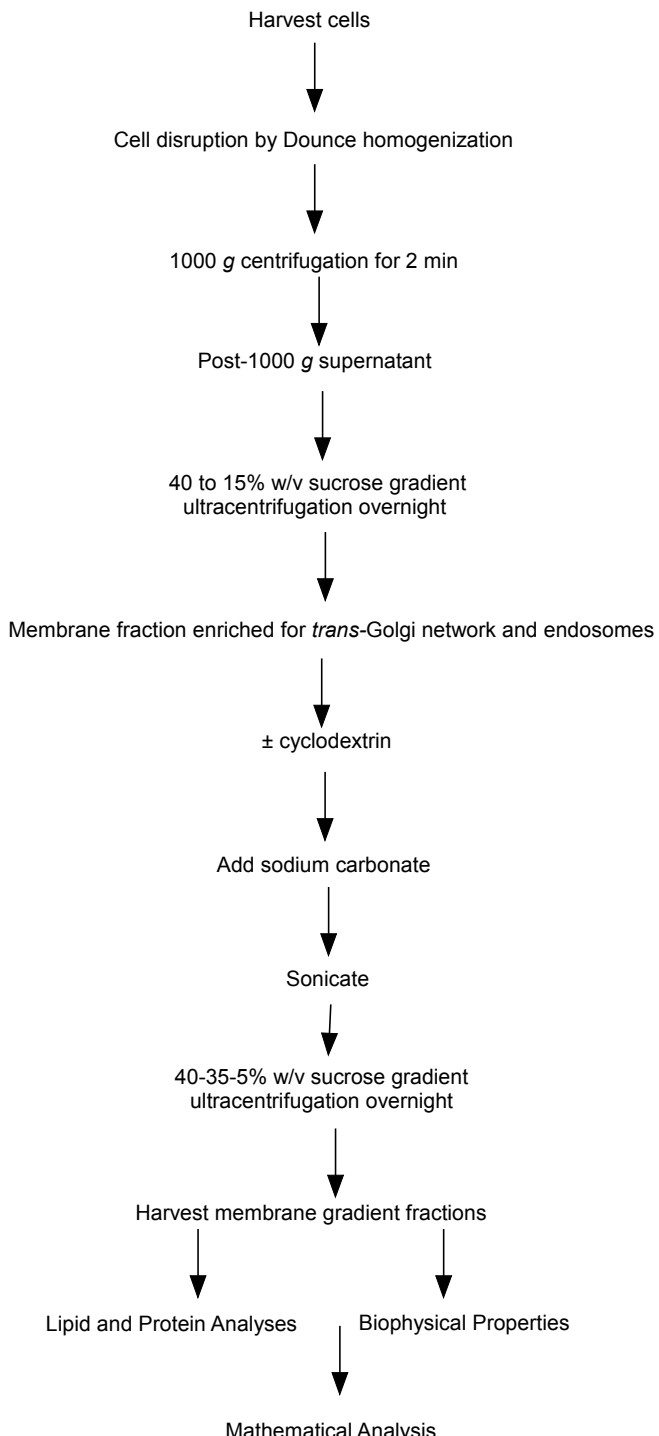

**Figure 1 Flow chart of steps involved in the subcellular fractionation procedures.** Flow chart outlining the steps involved in the subcellular fractionation procedures, equilibrium density floatation assay and membrane analyses used in the experiments.

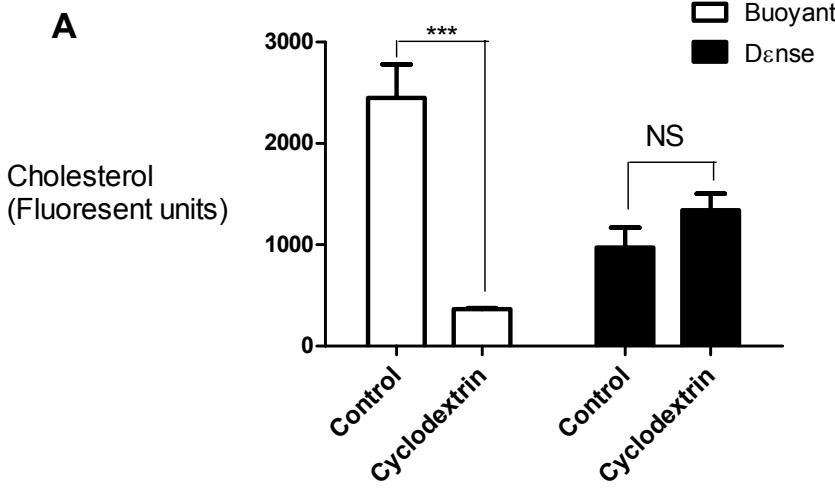

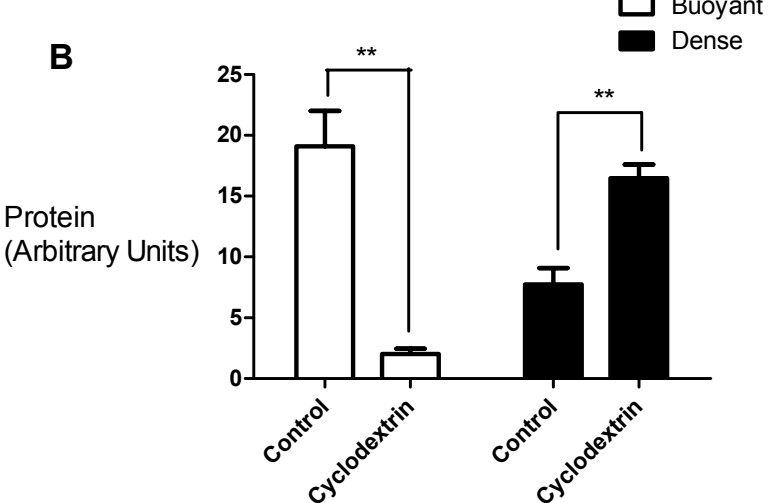

**Figure 2 Effects of cyclodextrin on vesicle composition.** Comparing the effects of cyclodextrin treatment on the biochemical composition of buoyant and dense membrane fractions isolated on equilibrium sucrose density gradients. (A) Change in cholesterol levels as determined by Amplex Red cholesterol assays. Note that there was no significant change in the total amount of cholesterol present in the dense membranes. (B) Levels of the membrane—associated protein PI4KIIα were determined by Western blotting and quantitated by image analysis software. Cyclodextrin addition causes a redistribution of PI4KIIα from the buoyant to the dense fractions. Results are presented as mean ± S.E.M from experiments repeated three times, ***$p < 0.001$, **$p < 0.01$, NS not statistically significant using the two-tailed student $t$-test.

brought about by cyclodextrin treatment caused the membrane vesicle sizes to decrease considerably.

Together, these experiments revealed that cholesterol depletion with cyclodextrin resulted in a reduction in membrane buoyancy, as evidenced by the delocalization of

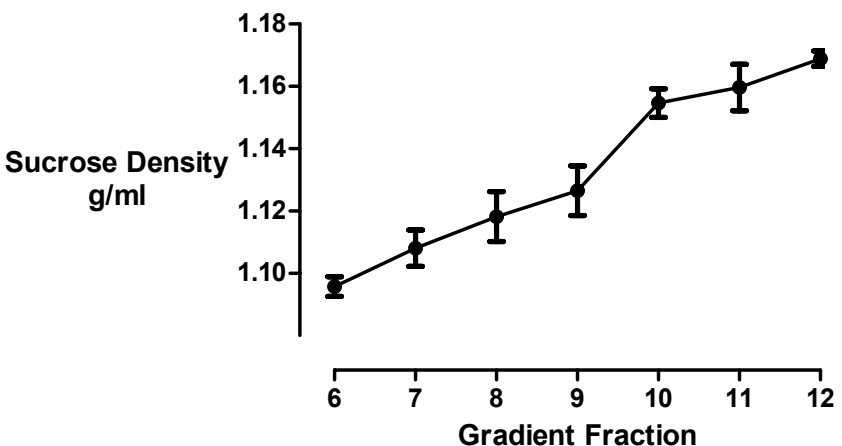

**Figure 3 Sucrose density gradient profile.** The density of each gradient fraction was determined by refractometry and the conversion of refractive index values to sucrose concentrations was accomplished using Blix tables. Results are presented as mean ± S.E.M of an experiment repeated three times.

**Table 1 Size of membrane vesicles in different gradient fractions following cholesterol depletion with cyclodextrin.** The size distributions, as measured by dynamic light scattering, of control and cyclodextrin-treated membrane vesicles from different gradient fractions. Results are presented as the mean ± S.D. of triplicate determinations.

| Treatment | Gradient fraction | Size (nm) |
|---|---|---|
| Control | Buoyant | 779.5 ± 28.2 |
| | Dense | 453.0 ± 177.1 |
| Cholesterol depletion | Buoyant | 42.2 ± 11.5 |
| | Dense | 270.2 ± 68.8 |

PI4KII$\alpha$-containing membranes to a denser region of the sucrose gradient and also a reduction in membrane size. Therefore, we decided to mathematically model the relationship between these different parameters.

## MATHEMATICAL MODELING

In this model, we determined an expected value for vesicle size post cyclodextrin treatment in our system and compared it with experimental data. From our experimental measurements, there was an 83.4 ± 2.75%, reduction in the cyclodextrin sensitive cholesterol component with other components not directly affected by the treatment. The total increase in mass density of the vesicles through cyclodextrin treatment was known (from 1.096 ± 0.003 mg/mL for fraction 5 to 1.122 ± 0.0005 mg/mL for fraction 10). The % composition of these two components and the density of the non-cholesterol residual component were unknown and approximated in this work, based on the above assumptions.

To determine the volumetric fractional composition of the vesicles pre cyclodextrin treatment and the density of the residual component, the experimentally measured values

were inserted into Eqs. (4), (6) and (7), giving Eqs. (8)–(10):

$$V_r \times \rho_r + V_s \times 1.067 = 1.096 \pm 0.003 \tag{8}$$

$$V_r + V_s = 1 \tag{9}$$

$$\frac{V_r \times \rho_r + V_s \times 1.067 \times (0.166 \pm 0.00275)}{V_r + V_s \times (0.166 \pm 0.00275)} = 1.122 \pm 0.0005. \tag{10}$$

Calculating these Eqs. (8)–(10) allowed us to predict the volume fractions of the vesicle pretreatment as follows—cholesterol $0.567 \pm 0.072$ ($56.7 \pm 7.2\%$), residual component $0.433 \pm 0.072$ ($43.3 \pm 7.2\%$), and a density of the residual component of $1.134 \pm 0.005$ mg/mL. As liquid-ordered domains typically comprises 20–30% cholesterol, the higher than expected value determined here is most likely the result of some membrane components being removed during the membrane isolation procedure and particularly by the alkaline carbonate addition step, leading to an apparent enrichment of cholesterol in the isolated fraction. Hence, the % value for cholesterol determined here is not the physiological proportion of cholesterol in TGN/endosomal membranes but rather, the amount present in the membrane vesicles after the extensive membrane disruption and isolation procedures used in this study. In concordance with this explanation, we have previously shown that membrane fractions prepared in the presence of carbonate are subject to substantial depletion of non-integral proteins (*Waugh, 2013*; *Waugh et al., 2011a*; *Waugh et al., 2011b*). As proteins have a density of 1.35 mg/mL (*Chick & Martin, 1913*; *Fischer, Polikarpov & Craievich, 2004*) and other membrane components such as lipids tend to have much lower densities, the value of $1.134 \pm 0.005$ mg/mL calculated for the density of the residual component seems reasonable.

Cyclodextrin treatment resulted in the total amount of cholesterol in the system to be reduced by 83.4%; however, the absolute volumes of the other components remained constant. To calculate the volume concentrations post cyclodextrin, three more simultaneous equations were formulated and solved by the same method:

$$V_{r\text{-post}} \times \rho_r + V_{s\text{-post}} \times 1.067 = 1.122 \pm 0.0005 \tag{11}$$

$$V_{r\text{-post}} + V_{s\text{-post}} = 1 \tag{12}$$

$$\frac{V_{r\text{-post}} \times \rho_r + V_{s\text{-post}} \times 1.067 \times 6.02}{V_{r\text{-post}} + V_{s\text{-post}} \times 6.02} = 1.096 \pm 0.03. \tag{13}$$

The predicted vesicle composition post treatment obtained by solving any two of Eqs. (11)–(13) was: cholesterol $0.179 \pm 0.047$ ($17.9 \pm 4.7\%$) and residual component $0.821 \pm 0.047$ ($82.1 \pm 4.7\%$).

The relative volume of the treated vesicles was calculated through Eq. (14):

$$V_r + V_s \times 0.166 \pm 0.0275 = \text{New Volume} \tag{14}$$

giving a relative volume of $0.527 \pm 0.073$, i.e., post treatment, the vesicle was $49.1 \pm 7.3\%$ of its original size. This corresponds to the diameter of the treated vesicles of $0.81 \pm 0.04$, i.e., the radius must have shrunk by $19 \pm 4\%$.

Experimental measurements showed the diameter of the vesicles falling from 453 ± 177.1 nm to 270.2 ± 68.8 nm post treatment, a 40.4% decline in diameter and thus an 88.8% fall in vesicle volume. This differs markedly from what has been calculated based on cyclodextrin affecting cholesterol alone and is consistent with previous work demonstrating that cyclodextrin can also sequester a range of hydrophobic molecules (reviewed in *Zidovetzki & Levitan, 2007*). These results imply that only about 50% of the change in membrane size is due to cholesterol desorption.

Since the mathematical analysis demonstrated that the decrease in membrane size could not be fully accounted for by cholesterol loss, we investigated the effect of cyclodextrin addition on the levels of membrane gangliosides which are glycosphingolipids that are structurally unrelated to sterols. Changes in ganglioside lipid distribution were determined using HRP-conjugated cholera toxin B subunit as a probe. Kuziemko and colleagues (*Kuziemko, Stroh & Stevens, 1996*) previously determined that Cholera-toxin binds to gangliosides in the order GM1 > GM2 > GD1A > GM3 > GT1B > GD1B > asialo-GM1, albeit with a >200 fold difference in binding affinity between GM1 and asialo-GM1. Therefore, unlike thin layer chromatography, dot-blotting immobilized sucrose density gradient fractions with cholera toxin B subunit does not permit the separation and quantitation of individual glycosphingolipid species. Furthermore there is a possibility that in addition to gangliosides, the toxin may also bind to the carbohydrate moieties of glycosylated proteins associated with the isolated vesicles (*Blank et al., 2007*; *Uesaka et al., 1994*). Bearing in mind these limitations, we used this well established technique (*Clarke, Ohanian & Ohanian, 2007*; *Correa et al., 2007*; *Domon et al., 2011*; *Ersek et al., 2015*; *Ilangumaran et al., 1996*; *Liu, Yao & Suzuki, 2013*; *Liu et al., 2015*; *Mazzone et al., 2006*; *Nguyen et al., 2007*; *Pang, Urquhart & Hooper, 2004*; *Pristera, Baker & Okuse, 2012*; *Russelakis-Carneiro et al., 2004*; *Tauzin et al., 2008*; *Waugh, 2013*; *Waugh et al., 2011a*; *Waugh et al., 2011b*) to generate a composite yet simple signal to assess if there was any redistribution of these structurally related non-sterol molecules in the density gradient following cyclodextrin addition (Fig. 4). We observed that the ganglioside content of the buoyant fractions was decreased by about 50% following cyclodextrin treatment and this is consistent with mathematical analysis that vesicle size reduction is due to the non-selective desorption of membrane lipids.

## DISCUSSION

Our combined biophysical, biochemical, and mathematical analyses demonstrate that cyclodextrin-induced cholesterol extraction can lead to an increase in equilibrium density by inducing membrane shrinkage. The cyclodextrin-induced shift of biomolecules to a denser membrane fraction can be accounted for by a large change in vesicle volume, without necessarily having to evoke the disruption of liquid-ordered membrane microdomains. These new findings have implications for the use of cyclodextrin-induced sterol depletion as a means of assessing whether a protein associates with cholesterol-rich lipid rafts. At high cholesterol levels, such as those reported here in the control buoyant membranes, one might expect significant levels of lipid rafts or even for the entire

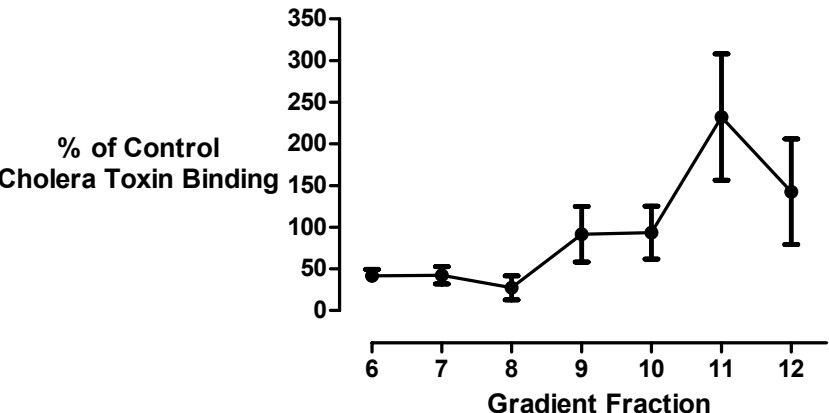

**Figure 4 Effect of cyclodextrin on ganglioside distribution profile.** Dot blotting of equal volume membrane fractions and detection with HRP-conjugated cholera toxin B subunit was used to determine the levels of ganglioside lipids in control and cyclodextrin-treated membrane fractions. Cyclodextrin addition resulted in a decrease in HRP-conjugated cholera toxin B subunit binding to the buoyant membrane fractions. Results are presented as mean ± S.E.M from experiments repeated three times.

membrane to exist solely in the liquid-ordered phase (*Almeida, Pokorny & Hinderliter, 2005*; *Armstrong et al., 2013*; *Munro, 2003*; *Swamy et al., 2006*) and hence, removal of cholesterol with cyclodextrin would be predicted to disrupt these rafts (*Cabrera-Poch et al., 2004*; *Kabouridis et al., 2000*; *Larbi et al., 2004*). However, in the context of the type of experiments described here, a cyclodextrin-dependent change in membrane density may only imply that a biomolecule is associated with a cholesterol-rich membrane and does not necessarily report the stable association of that component with lipid raft microdomains.

Our results suggest that at least under the experimental conditions employed here, cyclodextrin-induced reduction of membrane size can also be effected by the extraction of molecules other than sterols. The apparent lack of selectivity for cyclodextrin-induced biomolecule desorption demonstrated here leads us to speculate that these agents could potentially be repurposed to treat a range of conditions similar to Niemann-Pick type C, that feature enlarged endosomal membrane phenotypes due to defective lipid trafficking and/or metabolism but importantly, do not necessarily involve cholesterol accumulation. An example of a disease to consider in this regard could be oculocerebrorenal syndrome of Lowe (OCRL), a neurodevelopmental condition characterized by phosphatidylinositol 4,5-bisphosphate accumulation on endosomal membranes due to inactivating mutations in the OCRL phosphoinositide 5-phosphatase (reviewed in *Billcliff & Lowe, 2014*; *Clayton, Minogue & Waugh, 2013b*). Furthermore, whilst cyclodextrin has a high affinity for sterol lipids it is also known to bind phosphoinositides such as phosphatidylinositol 4-phosphate (*Davis, Perera & Boss, 2004*), and this further supports the idea that these macromolecules could have applications in the treatment of a number of inherited phospholipid storage disorders. This suggests a new type of drug action involving agents designed to alter membrane surface area through the reduction of membrane mass. The objective of such treatments would be to increase the membrane concentrations of more cyclodextrin-resistant biomolecules, in order to restore or amplify membrane-based

signaling or trafficking functions. This has already been shown for the epidermal growth factor receptor, which is subject to augmented levels of constitutive activation following cyclodextrin treatment (*Pike & Casey, 2002*; *Westover et al., 2003*). However, these possible uses for cyclodextrin remain speculative and further work is required to investigate if the biophysical changes documented here under specific in vitro conditions also occur on intracellular membranes in live cells.

In conclusion, this work throws new light on the mechanism of action of methyl-$\beta$-cyclodextrin on biological membranes. This may lead to a reassessment of its use in cell-based laboratory experiments while at the same time widening its potential applications in the therapeutic arena. In particular, this study indicates that the cholesterol-independent effects of cyclodextrin on membrane area may have more general applications in the treatment of intracellular lipid storage diseases.

### Nomenclature

$\rho$          mass density (kg/L)
$V$          volume (L)

### Funding

Financial support was provided by the Royal Free Charity (Dr. Mark G. Waugh), and the Royal Society for University Research Fellowship (Prof Nguyễn T.K. Thanh). The funders had no role in study design, data collection and analysis, decision to publish, or preparation of the manuscript.

### Grant Disclosures

The following grant information was disclosed by the authors:
Royal Free Charity.
Royal Society for University Research Fellowship.

### Competing Interests

The authors declare there are no competing interests.

### Author Contributions

- Peter Kilbride performed the experiments, analyzed the data, wrote the paper, reviewed drafts of the paper.
- Holly J. Woodward performed the experiments.
- Kuan Boone Tan performed the experiments, analyzed the data, contributed reagents/materials/analysis tools, reviewed drafts of the paper.
- Nguyễn T.K. Thanh analyzed the data, contributed reagents/materials/analysis tools, wrote the paper, reviewed drafts of the paper.
- K.M. Emily Chu performed the experiments, wrote the paper, reviewed drafts of the paper.

- Shane Minogue contributed reagents/materials/analysis tools, reviewed drafts of the paper.
- Mark G. Waugh conceived and designed the experiments, performed the experiments, analyzed the data, contributed reagents/materials/analysis tools, wrote the paper, prepared figures and/or tables, reviewed drafts of the paper.

### Supplemental Information

Supplemental information for this article can be found online at http://dx.doi.org/10.7717/peerj.1351#supplemental-information.

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
