# Peer review of "Modeling the effects of cyclodextrin on intracellular membrane vesicles from Cos-7 cells prepared by sonication and carbonate treatment"

_PeerJ, doi:10.7717/peerj.1351_

## Round 0.1 · original submission · Major Revisions

Please play particular attention to the comments from the reviewers which relate to having the appropriate controls for the experiments.
Note also the comments about the selectivity of cyclodextrin and possible overinterpreation of the results. Some of the discussion/interpretation could be softened or qualified further.

·

Basic reporting

The reported finding is that cyclodextrin treatment reduces membrane size and vesicles become denser, as assessed by a flotation assay and by hydrodynamic measurements.

Experimental design

The flotation assay is published elsewhere.
Several aspects of the mathematical treatment are described twice.

Validity of the findings

Results are possibly overinterpreted, in view of my comment to the authors (see below).

Additional comments

The work “Cholesterol depletion with cyclodextrin has major effects on membrane size”, by Peter Kilbride, Holly J. Woodward, Kuan Boone Tan, Nguyễn T.K. Thanh, K.M. Emily Chu, Shane Minogue, and Mark G Waugh explores the effect of methyl-beta-cyclodextrin (cyclodextrin)-mediated cholesterol extraction on the size and density of intracellular vesicles. The main finding is that cyclodextrin treatment reduces membrane size and vesicles become denser, as assessed by the flotation assay employed and by hydrodynamic measurements.
Cyclodextrins are mainly employed to modify cell-surface cholesterol in acutely treated cells. The choice of intracellular vesicles to test the effect of cyclodextrins on the physical properties is not properly explained, and appears somewhat arbitrary in view of the conclusions drawn by the authors on the implications of their findings in previous studies using acute cyclodextrin treatment to deplete cholesterol levels in cells and to disrupt lipid “rafts”.

The main difficulty with this work is that the methodology described under “Membrane floatation assay” involves not only 1) 10 mM cyclodextrin treatment of the TGN/endosomal subcellular fraction obtained from the Cos-7 cells, but in fact also comprises 2) high pH (pH 11.0 with 0.5 M carbonate) treatment and 3) physical disruption of the membranes by sonication. These methodological aspects have, in the view of this reviewer, two important implications:
A) The changes observed in the TGN/endosomal membrane are probably a consequence of these three (not mild at all!) treatments combined, not just cyclodextrin treatment as claimed by the authors. Cyclodextrin treatment alone is already complex and implicates not only cholesterol depletion, a usually assumed oversimplification, but several other changes in membrane structure and function. In the absence of a dissection of the respective contributions of each of these physical and chemical manipulations of the membrane, the main claim of this work is not warranted.
B) The authors make interesting comments on the possible therapeutic applications of cyclodextrins in some diseases. The experimental conditions used in this work with isolated subcellular fractions are unlikely to reproduce the mechanisms followed by cyclodextrins on intracellular membranes in intact living cells, and hence the interpretation of the mechanism of action of cyclodextrins in phospholipid storage diseases should be taken with caution.

Abstract
The Abstract does not mention that the work is about cyclodextrin treatment of a lipid-raft-enriched membrane fraction isolated from intracellular trans-Golgi network and endosomal membranes, obtained from Cos-7 cells, as referred to in the Introduction and Material and Methods sections, respectively.
The term “membrane contraction” is confusing, and probably not appropriate to describe the “non-selective desorption properties of cyclodextrins”.

Results
Explanation of the mathematical formulae in this section is redundant. The methodology employed in the mathematical treatment should be restricted to the Material and Methods section.

References
Some references (e.g. 19, 30, 31) lack volume, page, etc.

Reviewer 2 ·

Basic reporting

The manuscript is interesting and dealing with methodologies that require some further care.
The Authors should consider the following comments.

-Page 4, line 7. Cyclodextrin selectively….
Cyclodextrin remove many components and is not selective. Cholesterol is removed in larger quantity with respect to other complex lipids, probably due to its mobility from the intracellular space to the membrane and to the extracellular space. Please see Journal of Lipid Research (2003) 44, 2142-2151.

-Page 8, line 14.
Cholera toxin recognize primarily GM1 gangliosides and cannot be used to detect all t gangliosides. In addition, I do not believe that Cos 7 cells contain a lot of GM1.
GM1 identification cannot be done by dot-blotting, where the toxin can recognize any Neu5Ac-Gg4, or portion of it linked to a protein. The good procedure requires extracting gangliosides, to separate them by TLC and finally submit them to cholera toxin immunoblotting.

-Figure 2, A and B.
10 min, 10mM cyclodextrin treatment should remove about 75% of cholesterol. In figure it seems much less.
The total content of PI4KII in treated and control fraction seems different Reduced). What about

-the fractionation procedure to separate lipid domains has been established using cells. Its application to a TGN enriched fraction should be validated.
What about the distribution of glycerophospholipids, particularly of dipalmitoylphosphatidylcholine, into the 12 collected fractions? A detailed analyses of gangliosides should be also necessary (see the comment above).

-The separated vesicles are quite heterogeneous in diameter and by laser light scattering it is no easy to derive precise data. This must be commented.

Experimental design

The experimental design related to complex lipid analyses requires be perfected.
I do not comment the physicochemical and mathematic parts of the paper that are out of my expertise.

Validity of the findings

I explain in box 1

Reviewer 3 ·

Basic reporting

No Comments.

Experimental design

The experimental design has been adequately done except for not considering another set of control experiments as specified in my comments for the authors.

Validity of the findings

The findings are valid based on the experimental design as described in the manuscript.

Additional comments

The manuscript by Kilbride et al., describes a study re-examining the effects of cyclodextrin on membrane cholesterol depletion. It provides a fresh look at this frequently used approach in studies examining effects of cholesterol depletion on cell membranes and their constituents. Overall, this is a worthwhile study shedding a new light on possible consequences of the interaction between cyclodextrin and cellular membranes. It indicates that besides depleting membrane cholesterol and disrupting lipid rafts cyclodextrin apparently also affects membrane size and buoyancy, which potentially opens a way for medical applications of cyclodextrin. The data are clearly presented and the manuscript is well written. Nevertheless, I have several comments, which the authors should consider during the revision of their manuscript.

Major:
- Generation of membrane vesicles stripped of cyclodextrin-sensitive cytoskeletal proteins (p. 7-8) may not be without problems given that "membrane fractions prepared in the presence of carbonate are subject to substantial depletion of non-integral membrane proteins" (p. 16). Type II phosphatidylinositol 4-kinase is tightly membrane-associated peripheral membrane protein, which behaves as an integral membrane protein that can also be associated with the cytoskeleton (e.g. Xu et al.,Plant Cell 4(8): 941–951, 1992). Consequently, this raises a question about how accurate and representative is the protein content measurement using PI4KII.
- Following on the previous comment, could the reduction in membrane buoyancy be a consequence of removal of the cytoskeleton bound PI4KII upon cyclodextrin induced cholesterol depletion rather than the relocalization of PI4KII-containing membranes to a denser region of the sucrose gradient (Fig. 2)?
- How valid is the assumption that one of the vesicle components is not affected by cyclodextrin? Is there no cholesterol in these vesicles and why not?
- The authors are advised to include another vesicle preparation in their experiments by using cytochalasin and/or latrunculin to alter the organization of actin filaments prior to treating the membranes with cyclodextrin.

Minor:
- P. 5, 2nd paragraph (bottom): "... on the grounds that 'in the' absence ..." (space)
- P. 9, 2nd line (top): " ... of a particle is 'given' as ..."
- P. 10, 3rd line (top): delete additional "to cyclodextrin".

---

## Round 0.2 · Minor Revisions

Could you please make a brief qualifying comment around line 376 regarding the possibility of protein components as suggested by Reviewer 2?

Reviewer 2 ·

Basic reporting

see below

Experimental design

see below

Validity of the findings

see below

Additional comments

Lipid domains are very dynamic structures and it is very important to know what it occurs to their lipid composition and organization following cyclodextrin treatment. The ”fact “ is not that cyclodextrin “can also release other lipids…”. It removes other lipids.

They are over 40 years that I am studying membrane complex lipids, from their chemistry to their biological properties, and I know very well that there are many papers reporting an incorrect use of cholera toxin for ganglioside recognition. I did not reviewed these papers. Thus is not a matter to report 15t or more of these papers.
The Authors probably did not understood correctly my concern. In my comment I reported “….Neu5Ac-Gg4, or portion of it linked to a protein.“ The reply of the Author was only related to gangliosides and not to proteins. This is an important point because no total lipid analysis is reported to validate the procedure (see below).
In addition, great care is necessary to report a sequence of affinity values for glycosphingolipids, the binding constant depending by the type of support to which the lipid are associated. The Authors should also see Masserini et al. Biochemistry (1992) 31(8):2422-6, reporting the affinity of cholera toxin fucosyl-GM1.

I asked a validation procedure for the preparation of lipid rafts from TNG fraction and the Authors reply that this has been done in Minogue et al 2010. A lipid domain is characterized by cholesterol and sphingolipid enrichment with respect to glycerophospholipids and proteins. I read this paper and did not find any lipid characterization along the fractions.

The Authors report that “The primary purpose of this study was to determine using a mathematical model….”. I do not understand how the mathematical model can give information if the biochemistry methodology is not correct. First, it is necessary to be in front of a correct experimental approach and then the mathematical model can be determined.

Reviewer 3 ·

Basic reporting

No further comments.

Experimental design

No additional comments.

Validity of the findings

No further comments.

Additional comments

The changes and additions the authors made have substantially improved the quality of the manuscript. In addition, all my queries and comments have been adequately addressed. I recommend the revised manuscript for publication in its present form.

---

## Round 0.3 · accepted · Accept

The qualifications in the revised manuscript point to and qualify the assumptions made in developing the model.

[Note from Staff: Outside of the system, Reviewer 2 expressed concern regarding the previous decision of Minor Revisions. For the sake of a complete record, we reproduce below Dr Separovic's thoughts on this issue]:

"Thank you for passing on the comments from Rev 2 as I cannot see them on the article website.

The manuscript was originally assessed by three reviewers: 1 & 3 recommended major revision and Rev 2 was a ’reject’. The authors have carried out major revision and made a thorough attempt to answer all three reviewers. In my opinion, the authors answered all three well with the exception of a couple of points from Rev 2. However, despite the major revision, Rev 2 still gave the same recommendation of ‘reject’, which I consider too harsh and suggested that the authors make a minor revision to accommodate Rev 2.

Rev 2 notes that the gangliosides are not uniquely identified by the cholera toxin assay and Neu5Ac-Gg4, or a portion of it linked to a protein, would also be identified. The authors were asked to qualify their interpretation to note that this would influence their results, but nevertheless this does not dismiss their findings. The other comment about lipid rafts and domains is open to interpretation and we have fun at membrane biophysics meetings debating the definition and what constitutes one or the other. Rev 2 has a narrow definition of a lipid domain but I am of the opinion that a lipid domain is a more general term.

In my opinion, the paper is acceptable but in deference to Rev 2, I asked the authors to make a minor revision. I hope the authors choose to make the reviewers’ comments public so that we can share them more widely and the readers are alert to the healthy debate within our community.

I very much appreciate the contribution of Rev 2 and you are welcome to pass on my response and/or paste it together with the reviews of the article."